# 'Communication, that is the key': a qualitative investigation of how essential workers with COVID-19 responded to public health information

Mark Roe [1] , Conor Buggy,[1] Carolyn Ingram,[1] Mary Codd,[1] Claire Buckley,[2] Mary Archibald,[1] Natalia Rachwal,[1] Vicky Downey,[1] Yanbing Chen,[1] Penpatra Sripaiboonkij,[1] Anne Drummond,[1] Elizabeth Alvarez [3,4] Carla Perrotta [5]

## ABSTRACT

**Objectives** To understand how essential workers with confirmed infections responded to information on COVID-19.

**Design** Qualitative analysis of semistructured interviews conducted in collaboration with the national contact tracing management programme in Ireland.

**Setting** Semistructured interviews conducted via telephone and Zoom Meetings.

**Participants** 18 people in Ireland with laboratory confirmed SARS-CoV-2 infections using real-time PCR testing of oropharyngeal and nasopharyngeal swabs. All individuals were identified as part of workplace outbreaks defined as ≥2 individuals with epidemiologically linked infections.

**Results** A total of four high-order themes were identified: (1) accessing essential information early, (2) responses to emerging 'infodemic', (3) barriers to ongoing engagement and (4) communication strategies. Thirteen lower order or subthemes were identified and agreed on by the researchers.

**Conclusions** Our findings provide insights into how people infected with COVID-19 sought and processed related health information throughout the pandemic. We describe strategies used to navigate excessive and incomplete information and how perceptions of information providers evolve overtime. These results can inform future communication strategies on COVID-19.

## BACKGROUND

Sharing public health information is a crucial step in creating awareness of threats and protective strategies.[1] The COVID-19 pandemic is unique in two ways. First, it is the first pandemic where digital technologies and social media platforms are used to share information to keep people safe.[2] Second, although little information was available on the novel SARS-CoV-2 pathogen in January 2020, an 'infodemic' had emerged within 4 months, prompting global responses such

## STRENGTHS AND LIMITATIONS OF THIS STUDY

⇒ The qualitative design enabled the collection of data on the experiences and perceptions of people who had laboratory-confirmed COVID-19 infections.

⇒ Throughout the study, the researchers used several best-practice strategies to complete thematic analysis of data from semistructured interviews.

⇒ Given the lack of prior qualitative research on this topic and cohort, we used inductive reasoning approach, which enables us to avoid assumptions of prior research or frameworks.

⇒ Participants were selected from a subgroup of the population known as 'essential workers' that could not avail of 'remote working' throughout the pandemic making their views are critical to informing public health policies that cater for cohorts at increased risk of infection.

⇒ Our recruitment strategy had a low response rate perhaps indicating a lack of willingness in this cohort to engage with COVID-19 research or initiatives associated with official public health organisations.

as the WHO's Information Network for Epidemics.[3]

While acknowledging of wide-spread access to information on COVID-19, little is known about how people receive and process such content. This is especially true for groups at a higher risk of infection who are associated with super-spreading events (SSEs) which have a greater influence on the trajectory of epidemics.[4]

Prior research on COVID-19 information has explored the public's perceptions of communication strategies and their ability to correctly answer questions on COVID-19 epidemiology.[5 6] These offer little insight into how people who were most at risk of infection (eg, high risk workers) make sense of information on COVID-19 or how they

For numbered affiliations see end of article.

**Correspondence to**
Dr Mark Roe; mark.roe@ucd.ie



believe communication strategies could be improved in the future.

Using semistructured interviews, this study aims to investigate how essential workers with confirmed infections responded to information on COVID-19. This research could inform workplace-related public health communications as the current pandemic continues and future pandemics emerge.

## METHODS

This study represents a subanalysis of a preregistered study to investigate the behaviours and contextual factors associated with COVID-19 outbreaks and SSEs (OSF: https://osf.io/aeg74).

### Patient and public involvement

There was no patient or public involvement in setting the research agenda.

### Data collection methods

Semistructured interviews were conducted using a password-protected Zoom account via a secure login. Only the researcher and invited participant were admitted to the call. A transcript of the audio recording was generated and corrected following a playback of the original audio file, enabling the researchers to remove any identifiable information. Interviews lasted a combined total of 7.6 hour (average 25.2±12.5 min) generating 45 987 words. Each participant was assigned a unique identification number. After this process, recordings were deleted from all sources. Four researchers completed the interviews following a series of pilot interviews with fellow team members (NR=8; MR=6; MA=2; CI=2). All interviews were conducted from 22 July 2021 to 7 August 2021.

For context, as of 22 July 2021, Ireland reported 289 139 laboratory-confirmed infections since the first case on 29 February 2020, equating to a population prevalence of 5.8%.[7] The median age of national cases was 22. Healthcare workers accounted for 10.3%. A total of 15 134 hospital admissions and 5026 deaths were reported. Approximately, 63.8% and 53.0% of the population had been partially or fully vaccinated against COVID-19. In the 14 days prior to data collection, Ireland confirmed 13 065 infections (prevalence=0.3%) with 29.5% of cases associated with community transmission.[8] The median age was 23. A similar infection rate was reported until 13 October 2021. Ireland's 7-day test positivity rate was 5.9%–7.9% throughout the study period.[7]

We developed an interview guide to explore three areas: behaviours and contextual factors leading to COVID-19 outbreaks, responses to minimise transmission and lessons for future responses. These areas were identified following a review of the literature, consultation with national public health authorities and initial findings from a larger COVID-19 research project by the authors (Science Foundation Ireland: 20/COV/8539). Within this, we asked specific questions on information

and public health messaging and the role of the media in minimising transmission and sustaining safe behaviours. Prompt questions were asked by the researchers to illicit more detailed information on these topics.

### Study participants

We recruited 18 people in the Ireland who had laboratory-confirmed SARS-CoV-2 infections using real-time PCR testing of oropharyngeal and nasopharyngeal swabs. All individuals were identified as part of suspected workplace outbreaks, where ≥2 individuals had epidemiologically linked infections. We targeted this group given their first-hand experience of (1) COVID-19 outbreaks in the workplace and (2) responses to minimise transmission such as infection control measures in workplaces, engagement with test-and-trace services and quarantining. Therefore, inclusion criteria included age ≥18 years and a laboratory-confirmed infection likely linked to a workplace outbreak.

Potential participants were selected from a subgroup of the population known as 'essential workers' that could not avail of 'remote working' throughout the pandemic. Potential participants (n=167) were identified by the partner Health Service Executive (HSE) Contact Tracing Centre (CTC) as fulfilling the inclusion criteria, and during routine surveillance activities, participants were invited to receive a call from the research team. Overall, 104 individuals were contactable by the HSE CTC (62.3%). From this cohort, 64 (61.5%) refused to be contacted by the research team while 40 (38.5%) gave consent. We were able to contact and complete interviews with 18 people from this sample, representing 10.8% of potential participants originally identified by the HSE CTC.

Seven women (38.9%) and 11 men (61.1%) were interviewed. The median age was 35 years (min=18; max=57). Sectors represented included food, construction, laundry, retail, energy, healthcare, professional services, public services and emergency services.

### Data analysis

We used a thematic analysis approach to support a structured exploration of patterns in perceived behaviours and contextual factors associated with COVID-19 outbreaks experienced by the participants. As qualitative research seeks to generate knowledge grounded in human experience, thematic analysis is a trusted method that guides researchers in 'highlighting similarities and differences', 'exploring unanticipated insights' and patterns, and 'summarising key features' of data that cannot be statistically analysed.[9] Failure to overlook the importance of qualitative methods diminishes our understanding of how people experience specific events and how that experience might be enhanced through new considerations when designing interventions (eg, infection control measures in high-risk workplaces).

The coders (MR ×10, VD ×8) familiarised themselves with the transcripts prior to initiating coding. Thematic analysis was conducted in accordance with the six phases

of trustworthy thematic analysis as outlined by Nowell *et al* (ie, familiarisation with the data; generating initial codes; searching for themes; reviewing themes; defining and naming themes; producing the report).[9] Qualitative analysis software (NVivo V.12) was used to transform raw data into multiple codes using open and axial coding processes. A final of 35 codes were computed.

We considered saturation to be met when two conditions were realised: (1) agreement between researchers that no further transcript codes could be generated and (2) no further information, deemed relevant to the stated research questions, could be obtained from the transcripts.[10] Discrepancies were resolved through discussion until a consensus was reached. Reflective discussions were conducted (MR, VD, CI) throughout the study to review the rationale for selecting codes, themes and quotes.[11]

Credibility strategies used throughout analysis included (1) critical friend approach by encouraging reflexivity by asking provocative questions to critique our research in a supportive manner that clarifies ideas and interpretation and (2) cross-checking of themes identified in prior focus group studies.[12] The team has a diverse range of background (eg, medicine, public health, epidemiology, risk management, occupational health, psychology) that facilitated a robust review of identified themes. Results are reported in line with Standards for Reporting Qualitative Research (SRQR) (online supplemental file 1).[13]

## RESULTS

A total of four high-order themes were identified: (1) accessing essential information early, (2) responses to emerging infodemic, (3) barriers to ongoing engagement and (4) communication strategies. Thirteen lower order or subthemes were identified and agreed on by the researchers (figure 1).

### Theme 1: Accessing essential information early

The first theme highlights that participants reported access to information on the essential elements of COVID-19 before a pandemic was first declared in March 2020. The identified subthemes reveal the role of clear communications and personal thresholds for seeking information to reduce transmission of the virus in work settings.

### Subtheme 1.1: Clarity on what to know and do to prevent infection

Participants expressed the view that since COVID-19 was first confirmed in Ireland on 29 February 2020, public health information was almost unavoidable given the extent of coverage on news programmes and social media. Most participants accessed information via national television news shows which included live broadcasts of public health press briefings and newspapers as well as websites of the HSE and WHO. Additionally, every participant reported having access to information on COVID-19 in their workplace. Workplace information provided clarity on what staff could expect if workers developed symptoms of COVID-19. Interestingly, it appears that prior to their infection, participants largely tended to only seek generic information on COVID-19 that was deemed necessary for functional tasks (table 1).

### Subtheme 1.2: Personal scenarios triggering in-depth information seeking

Participants referred to triggers that initiated searches to answer specific queries related to COVID-19 that arose from unique personal events (table 1). This was largely indicative of personal concerns related to changes in their own health status, or that of an acquaintance or family member, leading to a reappraisal of the threat posed by COVID-19. More in-depth information was sought from specific Google searches, YouTube videos and formal research papers. It appears these intentional information seeking activities served the specific purpose of helping the participants make sense of their personal scenarios.

### Subtheme 1.3: Infection triggering clinical information seeking

We also noticed that even following confirmation of a positive test, participants only sought information when

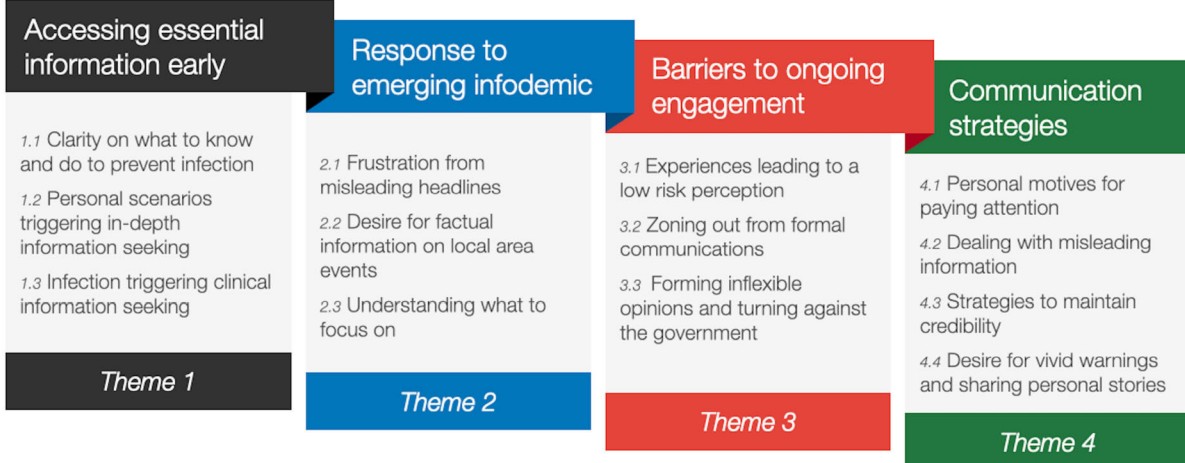

**Figure 1** Outline of higher order and lower themes on COVID-19 communication strategies. Uploaded separately as PDF as per editorial office request.

**Table 1** Illustrative quotes supporting theme one 'accessing essential information early'

| Sub-theme | Supporting quotes | Participant ID |
|---|---|---|
| 1.1 Clarity on what to know and do to prevent infection | You were kind of just told what the symptoms were, and you were educated on what symptoms to look out for and if you have any of these symptoms to report them and to isolate at home. | PID 12: Nurse |
| | Everyone was already pretty much informed of what would happen if there was a scenario where someone gets sick. | PID 17: Meat plant worker |
| | The procedures were rolled out nationally. There was no issue with dissemination of information. The quality of information coming at a national level, it was good. | PID 20: Emergency services worker |
| | There's a lot of access (to information) in every part of the company.(It's been)very very good through the COVID period so I give out credit for that. | PID 7: Food factory worker |
| | I got the information I needed, I never looked for any more information that I didn't need. | PID 12: Nurse |
| 1.2 Personal scenarios triggering in-depth information seeking | It's [not] until either they get COVID or they see someone that they know get COVID badly, once that happens then they'll realize what COVID is, and that's the sad reality is that there's a lot of people who won't take this thing seriously until they see somebody that they know with COVID. | PID 1: Meat plant worker |
| | When I got the headache I obviously was looking up to see was this part of the symptoms. Is it actually COVID or is it just this cough I had before Christmas? So I definitely was seeking out symptom information, and I read, I don't know it was like a dissertation or a there was a paper written on this 72 hour headache that wouldn't react to any medication or anything so that kind of rang true. | PID 16: Office worker in professional services |
| 1.3 Infection triggering clinical information seeking | I won't really be looking for information unless I like start showing symptoms and I start to feel very unwell then I'll start looking at it going like, right, what is it, how can I help myself with this, what can I do. | PID 12: Nurse |
| | [At the time] I was just kind of reading through different information about symptoms and how long they last and different things that can happen. | PID 16: Office worker in professional services |

COVID-19 began to impact their health (table 1). A clear distinction was noted between this and subtheme 1.2, as information seeking occurred after a specific event (ie, following confirmed infection) and refers to the specific domain of case management (ie, what are the potential clinical outcomes, and what actions can support better outcomes). Participants expressed that these searches for specific clinical information allowed them to prepare for what might be expected to occur throughout their period of self-isolation.

### Theme 2: Response to emerging infodemic
The second theme emerged in response to participants frequently discussing the rapid growth of information on COVID-19 despite it often being incomplete, and at times, misleading or of little use to inform their personal actions. Here, we noticed a desire for information based on how the COVID-19 epidemic was progressing in local areas.

### Subtheme 2.1: Frustration with misleading headlines
We repeatedly found references to perceived over-reporting on COVID-19 by the media (table 2). While participants mentioned that although the sustained media coverage was at times fear-provoking, it also played a useful role in keeping the public informed of the latest COVID-19 trajectory. Participants reported being disheartened by the manner of reporting, as opposed to the constant publication of media content related to COVID-19. Similarly, participants also alluded to their distaste for some marketing tactics, deployed by some

media outlets to increase online traffic, which had the potential to misinform the public.

### Subtheme 2.2: Desire for factual information on local area events
Our analysis identified that participants valued information related to their local geographical area, going beyond the reporting of mere confirmed case numbers to reveal details of infection sources and clinical outcomes (table 2). This was seen as beneficial to raising awareness of the threat in the area in attempts to bolster adherence to infection control measures.

### Subtheme 2.3: Understanding what to focus on
Despite repeated references to excessive amounts of information on COVID-19 in the media and online, participants appear to have coped by considering the relevance of the content for understanding the nature of the virus, and how it could inform their future actions (table 2).

### Theme 3: Barriers to ongoing engagement
Theme 3 may be best conceptualised as the consequences of participants having to seek and process information on COVID-19 in a largely ad hoc personally motivated manner (theme 1) while dealing with the factors of incomplete and context-free information (theme 2). We heard how this led some essential workers to reduce their perception on the threat of COVID-19 while simultaneously changing the nature of their engagement with official sources of public health information. Within this process, social media sites became platforms playing an ever-increasing role in reforming opinions on COVID-19

**Table 2** Illustrative quotes supporting theme two 'response to emerging infodemic'

| Subtheme | Supporting quotes | Participant ID |
|---|---|---|
| 2.1 Frustration with misleading headlines | I feel like the cases are kind of annoying hearing about [them] every day, but I feel like it is good to know… I think they just report on it too often kind of scaring people like as in they kind of use headlines to scare people, when you read the actual information it's very misleading. | PID 14: Factory worker |
| | Give proper factual information instead of kind of click bait headlines on everything. | PID 14: Factory worker |
| 2.2 Desire for factual information on local area events | There should be maybe more information on it like how many people are like in hospital in each area and like where the cases are explicitly coming from.(I)think it'd be more beneficial than maybe just hearing the cases every day, you kind of stop listening and stop caring, there's like 1000 cases when like 4 million people like it doesn't, but if you kind of read it and know people are near you it might make people pay more attention and stick to the rules and keep safe | PID 14: Factory worker |
| | I think the government needs to be more transparent on things and provide more accurate information because at the time when this whole pandemic was rampant I feel like the government was just overloading people with so much information that you didn't even know which one is right which one is wrong. You could be saying 'oh that's what I heard is' and then another person could be saying 'oh no, this is what I heard is' and like you know it's all a bunch of everything. All the information is all jumbled up and the truth is somewhere in the middle. | PID 17: Meat plant worker |
| 2.3 Understanding what to focus on | It's just knowing what this virus is and knowing how it's going to affect your body and being able to prepare yourself for when it's going to affect your body | PID 1: Meat plant worker |
| | I was just reading more about it and what it actually does and how or what I'm supposed to take or do to make my immune system stronger, so those were the things I was checking | PID 3: Laundry worker |
| | I guess the main thing is have information readily available for everybody just to keep on track and keep updated on what the most recent guidelines are. | PID 12: Nurse |

as well as the government and officials responding to it. This was rooted in a perceived mismatch between information reported on COVID-19 and insights gained from first-hand experience of being infected, resulting in some participants stating that they will be less likely to engage with communications in the future.

### Subtheme 3.1: Experiences leading to a low-risk perception

We noted that several participants, having reflected on how the congruency between their personal experiences throughout the pandemic and those reported by the media, expressed a current belief that COVID-19 was not as dangerous as reported in the media (table 3). These participants believed that they correctly understood the content of prior media reports, yet witnessed, or heard about, contradictory COVID-19 outcomes prior to adopting their alternative beliefs. A source of evidence for personal experiences leading to a lower risk perception appears to be perceived wrong information that was reported yet never acknowledged or corrected. We noted that such participants were eager to emphasise that they were not denying the existence of COVID-19.

Of 18 participants, we encountered only one that alluded to conspiracy theories. Thus, we draw a clear distinction between the emergence of alternative hypotheses, at odds with the narratives reported by official authorities or media outlets, and conspiracy theories citing malicious motives by the official organisations.

### Subtheme 3.2: Zoning out from formal communications

Participants referred to a reduced ability to continually engage with media and public health messages over time, resulting in unintended outcomes such as when some people become 'accustomed to hearing about it and stop caring'. Formal communications strategies (ie, content released by national health authorities such as press briefings) on COVID-19 in Ireland were first implemented in February 2020. We noted the belief that the public developed a reduced willingness, or capacity, to engage in this form of messaging as the pandemic progressed (table 3). Instead, participants mentioned the role of 'informal advertisements', associated with the placing of public health messages during the start and end of popular programmes and delivered by the host.

### Subtheme 3.3: Forming inflexible opinions and turning against the government

Participants recognised the role of social networks, particularly those online, in shaping peoples' interpretation of information related to COVID-19. Specifically, we noted the perception that online social networks were effective at facilitating a process, whereby users arrived at an interpretation of COVID-19 information that was more likely to be anti-establishment (table 3).

Participants spoke of the consequences of prior measures taken to reduce transmission of COVID-19. In particular, national lockdowns, known locally as 'Level Five restrictions', were identified as being a source of anger among the public. We found that participants were unlikely to dissociate government officials from senior members of Ireland's National Public Health Emergency Team. Additionally, there was a sentiment that the public's negative reaction to previous COVID-19 responses would lead some people to adopt a negative attitude of future messages regardless of the content (table 3).

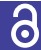

**Table 3** Illustrative quotes supporting theme three 'barriers to ongoing engagement'

| Subtheme | Supporting quotes | Participant ID |
|---|---|---|
| 3.1 Experiences leading to a low risk perception | I believe soon it's coming, another lockdown. Soon, very soon I believe, and I think that you know that already. There are so many cases that's why. Because I said like I don't know how [they're] can be so many cases. But we never… heard, like we saw on the news there are few cases like 900 or something like that a day… but we didn't hear like somebody had for real COVID. | PID 22: Meat plant worker |
| | It's not like we don't believe about COVID, alright it's a virus I understand, but it's not as worse [as] they actually say… How can you believe everything they say on the news and everything when people every day see different? | PID 21: Construction worker |
| 3.2 Zoning out from formal communications | So the only information I really looked for was [when symptomatic], I was talking to my boss at the time so he was the one who was keeping me informed with what I should be doing at all, so uh he told me to stay off. I think it was 2 days and then if I still had symptoms or if symptoms [hadn't] got better I was to get a test at the GP. | PID 17: Meat plant worker |
| | I feel like some people when they kind of read the news like what will make them be more careful, but sometimes it's the opposite. Like if people are hearing it every day they just kind of just become accustomed to hearing about it and stop caring. | PID 13: Nurse |
| | You always hear [TV host] when he's on(Ireland's most popular entertainment show)wrapping it up saying "everyone make sure to wear your masks and wash your hands", so if it was kind of embedded into more of an informal advertisement. I think it could be responded to better or even like on the radio channels just that it's nearly embedded subconsciously into people. But I think the formality on how they're doing it now, it's just lost the interest of people. People are just zoned out they don't care anymore about it. | PID 13: Nurse |
| 3.3 Forming inflexible opinions and turning against the government | People are very headstrong. They'll keep this thought that they have of whatever their opinion on COVID is because, like, they'll have one or two people agree with them on Facebook or something, and they'll think alright well what I'm saying is correct. | PID 1: Meat plant worker |
| | Some people like they read too much online on Facebook, on whatever like Instagram or whatever. And then [get] sucked into it and then they end up believing it, and then it just keeps going on and on until they're sitting there telling people that COVID isn't real or that the vaccine is trying to control them. But, like, it's just that some people think that anything they're told shouldn't be taken at face value. Some people find it hard that when the government says something, that's the truth, some people do find that very hard. | PID 1: Meat plant worker |
| | The main thing is, especially with the measures that we had in place when we were in Level Five, I think those were very, they were very harsh. I understand. Yet it did work. But it didn't really. Like, there were a lot of people that turned against the government. | PID 1: Meat plant worker |
| | I think to be honest with ya a lot of people have a negative response to the government and most of the directors I've heard from the government. I know a lot of people don't like [Ireland's Chief Medical Officer]. I don't mind him, but a lot of people don't like him, and if they say something they're [of] the attitude that they're not just going to take it on board, they're just going to be disrespectful to him and it's just going to ignored from the get go. | PID 13: Nurse |

### Theme 4: Communication strategies

The final theme we identified related to how participants believed that future communication strategies on COVID-19 and future public health emergencies could be improved.

### Subtheme 4.1: Personal motives for paying attention

Despite referring to excessive information and media coverage, all participants appeared to value ongoing communication. Similar to our findings in subtheme 1.2 (ie, personal scenarios triggering in-depth information seeking), participants expressed a need for future communications to be anchored to a personal value proposition, such as protecting loved ones (table 4).

### Subtheme 4.2: Dealing with misleading information

All participants used social media yet were unclear about specific measures that could be taken to reduce the misinformation on digital platforms (table 4).

### Subtheme 4.3: Strategies to maintain credibility

Our analysis indicates that to avoid misinformation, participants adopted a strategy of seeking information from several recognised health authorities, rather than a sole source, before cross-checking the content with media reports (table 4).

### Subtheme 4.4: Desire for vivid warnings and sharing personal stories

When considering lessons for future strategies to improve adherence to public health measures, participants again referred to personal scenarios. Specifically, strategies used to respond to promote awareness of other public health threats, such as smoking, were mentioned by participants (table 4).

### DISCUSSION

Our study investigated how essential workers with confirmed infections responded to public health

**Table 4** Illustrative quotes supporting theme four 'communication strategies'

| Subtheme | Supporting quotes | Participant ID |
|---|---|---|
| 4.1 Personal motives for paying attention | Communication that's the key to explain to them, to explain things to them because if you love someone, your parent, if your parent is vulnerable, you need to protect them, we protect each other… if I protect myself, I protect also my family… If we pay attention [to] the information, we can protect each other. | PID 4: Laundry worker |
| | The information being disseminated is continued and is of good quality but that depends on the individual person, whether they decide to look at that information or read it. There was no onus on anybody to read it and if they didn't it wouldn't make a whole lot of difference to be honest with ya. | PID 20: Emergency services worker |
| | I think some people, they should get more information when we know [that COVID-19] is really happening in the world. So me and my daughter and my family we got [the] most information we could. But all the time we leave we put mask on, alcohol gel we have in handbag, all the time using sanitizer to clean the doors and stairs and everything. But when I start work, I see some people they doesn't care so much, you know. | PID 7: Food processing plant general manager |
| | I mean one key piece of information that I'd say is make sure people know that in the case of testing positive they won't go without pay. Because like I can see that causing a lot of unrest with people because you know it's like I said, a lot of people have families to feed you know and people rely on them for the money to get by. If people knew that and it's like 'oh I'll be taken care of if this happens' you know, they'll be a lot more willing to cooperate with the normal guidelines. | PID 17: Meat plant worker |
| 4.2 Dealing with misleading information | Maybe if there was something in place to kind of stop all the false news and kind of stuff that's spread online. | PID 14: Factory worker |
| | [When tested positive] the HSE [national health service] send me email for ten days about what I do [and to ask] how I feel. | PID 9: Meatplant worker |
| 4.3 Strategies to maintain credibility | All the guys are doing everything that [they] can do because there's so much more information about this… Because everything we know we get it from HSE [Health Service Executive], we get it from news, we get it from TV. Like everything, we get it from information from the health organisations giving to us. | PID 7: Food processing plant general operations |
| | It's just about reinforcing the information, the guidelines. It's a bit like any other training that we do in healthcare when it comes to basic life support or infection control or anything like that. We do annual updates of that every year and that is the simple purpose to keep the practice up, to keep your skills up and to keep it on the forefront of your mind so that you don't kinda forget and let standards slip. I think that's really the best way really to go forward, is to have support if you have any questions and have that support readily available but as well as that just having the information there and having regular updates and what's best to do. | PID 12: Nurse, female |
| | [When infected] I had access from the [contact tracing caller] to all the information. I feel like they had all the information. | PID 8: Meatplant worker |
| 4.4 Desire for vivid warnings and sharing personal stories | They should have a picture of what COVID-19 does to your body and they'll help people realize because, like, on the smoking packets, it does help people realise and they say on them like the dangers of smoking. It shows the harsh reality of it, and that's what people need to realise is the harsh reality. | PID 1: Meat plant worker |
| | From a public point of view, all I can say is again information. Just that information is put out there readily for everyone. That it's clear, it's concise, it's honest, and it's blunt with regards to the impact that it will have and what's expected. | PID 12: Nurse |
| | Like I got infected. I could tell them, 'this is my story and yes it [is] real. You should believe this'. | PID 3: Laundry worker |
| | Thank you for calling and thank you for letting me explain my experience because it's very hard when you are alone. Like I'm single mother, just me and my daughter, so if something happen with me I was scared with who was going to look after my baby and all this things. But at same time [being infected] make me more strong, make me look ahead and say listen 'this is going to pass, this is not going to affect you forever, this is just a moment, uh, so keep calm'. | PID 7: Food processing plant general operations |

information on COVID-19. Analysis of semistructured interviews with 18 participants identified four themes: (1) accessing essential information early, (2) response to emerging infodemic, (3) barriers to ongoing engagement and (4) communication strategies. No hypothesis was tested. Instead, we used inductive reasoning to explore patterns in COVID-19 information usage, identifying

several findings that can inform public health information strategies.

The precursor for seeking information is a perceived need to understand (ie, describe what is) and respond (ie, prescribe what should be done) to an unfolding event (ie, COVID-19).[14] These needs evolve overtime, are directed by personal experiences and determine the perceived

importance of seeking the best possible information or making correct decisions.[15]

Given the extent of coverage in the media, information on COVID-19 was practically unavoidable in 2020. Early information on COVID-19 appears to have had the effect of focusing attention on symptoms and how suspected cases should be managed. These early 'fundamental truths', even amid incomplete information, were simple to process and capable of informing public behaviours.[16] This is important as most of the public in the region of study has had no prior experience of pandemic pathogens.

We identified that most participants obtained information from official organisations such as national and international health organisations, indicating that the people interviewed in the current study saw these sources as credible, or at least, worth seeking guidance from. In some instances, failure to share the best available information led people to seek info from misleading sources as described in subtheme 2.2 *desire for factual information on local area events,* and 3.2 *zoning out from formal communications.* Disengagement is happening and can be problematic. Yet as it happens, we see examples of lived experiences contributing to healthy behaviours. Thus, a solution might be to facilitate exchanges between producers of information and the general population to (1) ensure that needs and population expertise are being heard and (2) encourage engagement by building trust through dialogue and providing ongoing explanation.

Prior studies provide evidence that the perceived risk of infection depends on the proportion of neighbours that are ill, meaning that amid a lack of official communication on local events, people turn to personal networks to gain information on their neighbourhood.[17] In the era of digital networks, this inevitably funnels people towards social media platforms, increasing exposure to misinformation and polarising opinions.[18]

We note calls for social media companies to regulate information deemed inaccurate, false or malicious.[19] However, that is unlikely to satisfy information needs as people will most likely default to word of mouth.[20] For instance, Lupton and Lewis noted that the public use informal communications with friends and families, local and internationally, to keep up to date with emerging information.[21]

While the role of clickbait headlines and social media is undoubtedly playing a role in spreading misinformation, so too is the inability of public health authorities to collect and disseminate information to people on live outbreaks in their communities. Surveillance capacities are a key component of the International Health Regulations for preparing and responding to infectious disease threats.[22] Tools such as community outbreak alerts might be particularly useful, especially as attention to specific diseases may decline after surges of infection when relatively low national figures are communicated.[17]

Although tools such as community outbreak alerts are common in many jurisdictions, Irish health authorities have not used them citing concerns related to data privacy; however, guidance from the national Data Protection Commissioner on 6 March 2020 states that 'Article 9 (2) (1) of General Data Protection Regulations and Section 53 of the Data Protection Act 2018 will permit the processing of personal data, including health data, once suitable safeguards are implemented. Such safeguards may include limitation on access to the data, strict time limits for erasure, and other measures such as adequate staff training to protect the data protection rights of individuals'.[23]

It is foreseeable that people receiving information at odds with their personal experiences are more likely to doubt the credibility of the sender(s). Our results imply that overtime, participants adopted a cautious attitude or scepticism towards information shared by official and government organisations, indicating a degree of frustration and mistrust.[24]

Gaps in information lead to more reliance on experiential knowledge, a recognised useful resource when responding to uncertainty (eg, learning from community leaders or professional mentors), especially if derived in settings that are difficult for outsiders to grasp.[25 26] This is likely attributable to concerns about COVID-19, and the response to it, 'bubbling up' slowly overtime as the pandemic progresses.[27] Factors contributing to these changes in personal risk perceptions included delays in collecting and reporting information, the presence of random noise in that information, and the shared belief in social networks that the posed risk is reducing over time.[28] Hence, we observed an interplay between people making an effort to stay informed while simultaneously withdrawing or disengaging from previously trusted sources, leading to emergence of more settled, and potentially alternative, beliefs running counter to official interpretations.[24] We believe that this is also evident in our ability to only reach 10.8% of our identified sample.

Adopting beliefs by processing imperfect information through social interactions indicates the use of well-documented heuristic principles, mainly the representativeness heuristic (ie, comparing risk situations with those that seem similar), the availability heuristic (ie, assessing risk with information most easily accessed) and the anchoring heuristic (ie, risk estimate based on previous event).[15 28] While this may seem undesirable to evidence-based domains, heuristics and risk management principles continue to form the basis of many effective COVID-19 measures that are defendable even in the absence of scientific evidence.[29 30]

Additionally, the true risk of COVID-19 events has been difficult to accurately assess and provide information on. For example, as evidenced by large differences between reported case numbers and seroprevalence studies.[31] Overtime, as dissatisfaction manifests and trust in the information sender is eroded, it is unlikely that responses such fact checking will deliver an immediate change in perception or behaviour.[32] In line with the action-based model of cognitive-dissonance processes, the sensation leading to a 'zoning out', signals a distancing from

pretrusted sources, as inconsistencies in information and cognition must be resolved through alternatives means for the person to take a 'logical course' of action.[33]

Hence, it is over simplistic to label 'zoning out' as a symptom of a catch-all term like pandemic fatigue (ie, a perceived exhaustion of cognitive and emotional capacities to sustain the effort required to positively respond to outbreaks), given that it is influenced by a complex two-way sender–receiver process of detecting, disseminating and deciphering key data, including outcomes of policies adopted by public health officials and government. Instead, it is perhaps more accurate to consider 'zoning out' as a stage in response to incomplete information from formal releases from state agencies that leads people to move away from previously reliable sources to seek additional information.

Participants referenced previous public health campaigns that involved graphic health warnings on cigarette packaging. Research has shown that the public has a preference for larger, pictorial and loss-framed warning labels given that these are more likely to attract attention and further thought or information seeking on health risks.[34] Such communication strategies have been shown to be particularly effective at generating motivation to change behaviour in cohorts aged 13–30 years; a group more likely to use online social media.[35 36]

Ultimately, this emphasises the importance of message content (ie, what to say) and message executions (ie, how to say it) when communicating information on COVID-19.[37] The need for reiterative approaches to develop public health communications has been highlighted and largely involves monitoring perceptions of message receivers of classic message inputs such as who, says what, through which channel, to whom, and with what effect.[37 38] However, regardless of the message executions or framing, our study found that interviewed participants were sceptical about the underlying assumptions of message content (eg, accuracy of information being communicated or the relevance to their local community).

We also note similarities between our findings on communication strategies and those from a thematic analysis of social media posts linked to a video-based quit smoking campaign that exposed the public to comments deemed both oppositional (eg, role of government, bigger perceived threats, scientific scepticism) and supportive (eg, personal stories, support for change, reactions to add content).[39] Critics of graphic health warnings often believe that these strategies are emotional and not factual; however, evidence suggests that there is little difference in emotional responses to information presented in graphic or factual text form, suggesting a false dichotomy.[36]

While sustained communication can overwhelm, it is also a valued strategy to alleviate anxieties and uncertainty. For messages to provide clarity, they must recognise both the change in information and misinformation.[21] This is because the public recognise that COVID-19 misinformation is common and expect scientists and governments to provide accurate information.[21]

Given the ever-evolving nature of infectious pathogens, the rapid detection and dissemination of health threats are heavily reliant on information technology (IT) infrastructure for two-way communication between health providers and the public.[22 24]

We also note references to sharing stories by participants. Like Bennett *et al* we believe that future communication strategies could learn from personal experiences of people infected with COVID-19 if given 'an outlet (to hear their) uncensored stories'.[40]

## Strengths and limitations

Our novel study design involved a partnership with the national contact tracing management programme. The qualitative design enabled the collection of data on the experiences and perceptions of people who had laboratory-confirmed COVID-19 infections. Throughout the study, the researchers used several best-practice strategies to complete thematic analysis of data from semistructured interviews. Additionally, given the lack of prior qualitative research on this topic and cohort, we used inductive reasoning approach, which enables us to avoid assumptions of prior research or frameworks.

A strength of our approach is the focus on essential workers given their repeated exposure to higher risk settings throughout the pandemic, particularly when stay at home orders were imposed for the rest of society. This allows us to capture unique experiences of essential workers who were at the coalface of dealing with the threat of COVID-19 before widespread vaccination in society. It is likely that these shaped participants' perceptions of national responses to COVID-19 as outlined in theme three.

A potential limitation of this study is that our recruitment strategy had a low response rate perhaps indicating a low willingness in this cohort to engage with COVID-19 research or initiatives associated with official public health organisations. This may bias our sample and could be addressed through future contact tracing strategies. Additionally, our sample size (n=18) means that our results cannot be generalised to the population or workforce as a whole. However, as participants represented essential workers, awareness of their views is important to inform public health policies that cater for cohorts at increased risk of infection.

## Author affiliations
[1]School of Public Health, Physiotherapy, and Sports Science, University College Dublin, Dublin, Ireland
[2]Health Service Executive Contact Management Programme, Health Service Executive, Dublin, Ireland
[3]Health Research Methods, Evidence and Impact, McMaster University, Hamilton, Ontario, Canada
[4]Centre for Health Economics and Policy Analysis (CHEPA), McMaster University, Hamilton, Ontario, Canada
[5]School of Public Health, Physiotherapy and Sport Sciences, University College Dublin, Dublin, Ireland

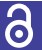

**Correction notice** This article has been corrected since it published online to reflect the correct name for author Natalia Rachwal.

**Contributors** MR, CI, MA, NR, CP, MC, CoB, CIB, YC, PS, AD and EA conceptualised and designed the study, and reviewed and revised the manuscript. MR, CI, MA, NR, CP, MC, CoB and CIB designed the data collection instruments and recruited participants. MR, CI, MA and NR conducted qualitative interviews. MR, VD and CI analysed qualitative data. MR and CI drafted the initial manuscript. All authors approved the final manuscript as submitted and agree to be accountable for all aspects of the work. MR is the author acting guarantor.

**Funding** This work was supported by Science Foundation Ireland (SFI) through the COVID-19 Rapid Response Research and Innovation Programme, grant number 20/COV/8539.

**Competing interests** None declared.

**Patient and public involvement** Patients and/or the public were not involved in the design, or conduct, or reporting, or dissemination plans of this research.

**Patient consent for publication** Not applicable.

**Ethics approval** Ethical approval was granted by the UCD Human Research and Ethics Committee (ID: LS-E-21-85-Roe-Perrotta). Verbal informed consent was obtained from the study participants.

**Provenance and peer review** Not commissioned; externally peer reviewed.

**Data availability statement** Data are available upon reasonable request.

**ORCID iDs**
Mark Roe http://orcid.org/0000-0001-6615-2283
Elizabeth Alvarez http://orcid.org/0000-0003-2333-0144
Carla Perrotta http://orcid.org/0000-0001-5986-4581

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
