## [Reviewer comments · BMJ Open]

ARTICLE DETAILS

TITLE (PROVISIONAL)	"Communication, that's the key": A qualitative investigation of how essential workers with COVID-19 responded to public health information
AUTHORS	Roe, Mark; Buggy, Conor; Ingram, Carolyn; Codd, Mary; Buckley, Claire; Archibald, Mary; Rachwal, Natalie; Downey, Vicky; Chen, Yanbing; Sripaiboonkji, Penpatra; Drummond, Anne; Alvarez, Elizabeth; Perrotta, Carla

VERSION 1 – REVIEW

REVIEWER	Sleigh, Joanna ETH Zürich, Health Sciences & Technology
REVIEW RETURNED	23-Feb-2022

GENERAL COMMENTS	This paper presents a qualitative analysis of structured interviews with 18 essential workers in Ireland with Covid-19 infections. Overall, I found this to be a well written paper which is suitable for publication with only some relatively minor issues to address: 1) P6, line 36. This sentence seems to be unfinished or misplaced.2) P7, line 35. How did you derive these topics for your interview guide? Was it based on previous literature?3) P7, Study Participants. I would suggest moving your explanation of 'essential workers' from page 20, line 46-49, to this section to improve clarity.4) P9, line 5. Could you please clarify what the 'critical friend' approach is and reference this method of credibility.5) P18, Discussion. While the aims and methods of the paper are clear and relevant, the paper doesn't quite draw on the significance of the interviewee subjects being essential workers. Is it possible to say a little more in the discussion about the significance of this.6) P18, line 60. "indicating people trust these sources" - this seems a bit of an overstatement. A sample of 18 cannot be generalised to 'people'.7) P19, line 3. Could you explain more fully by what you mean by: "In some instances, failure to share the best available information led people to seek info from misleading sources." I could not locate this in your results?
--

	8) P21, line 36-47. You mention the use of loss-framed messages as a means to motivate change in behaviour. However, message framing is a much debated topic. Your discussion would be strengthened here by linking to literature on the use of message framing during Covid-19. 9) Strengths and limitations: Please add that the small sample size means that the results are not generalisable.
--	--

REVIEWER	Al-Ghunaim, Tmam University of Leeds, school of psychology
REVIEW RETURNED	04-Mar-2022

GENERAL COMMENTS	I have some concerns about the article's first one in the introduction section; it is not clear for the reader what the aim of this study is? What does this study try to solve? Also, I have some concerns about using qualitative methods; why did you choose the interview rather than collecting data by the survey? In the method section, who is the target sample? I think this also you need to make it clear in the title. In the result section, We need to clarify the meaning of the theme before going to the subtheme. As a reader, it is unclear what you mean by barriers to ongoing engagement. Also, what is the link between the theme and the aim of the study? I think you need to link them in the result section.
--

VERSION 1 – AUTHOR RESPONSE

Response to Reviewer 1:

“Communication, that’s the key”: A qualitative investigation of how essential workers with COVID-19 responded to public health information

Dear Dr Joanna Sleight

On behalf of the authors I wish to extend my gratitude for taking the time to review our manuscript and making suggestions to clarify our findings. We appreciate your positive feedback on the importance and clarity of our findings and are pleased that our interpretation of the results aligns with external peer reviewers.

We have considered all of your comments and correct all unfinished sentences and expanded on the relevance of framing messages as it relates to our study. Below we outline our amendments to each of your individual comments.

We hope you that find now our manuscript ready for publication in BMJ Open.

Many thanks,

Mark

P6, line 36. This sentence seems to be unfinished or misplaced.

Correct. We have updates this sentence to read: “Using semi-structured interviews, this study aims to investigate how essential workers with confirmed infections responded to information on COVID-19.”

P7, line 35. How did you derive these topics for your interview guide? Was it based on previous literature?

We have added: “These areas were identified following a review of the literature, consultation with national public health authorities, and initial findings from a larger COVID-19 research project by the authors (Science Foundation Ireland: 20/COV/8539).”

P7, Study Participants. I would suggest moving your explanation of 'essential workers' from page 20, line 46-49, to this section to improve clarity. Thank you for highlighting this. We have moved this sentence.

P9, line 5. Could you please clarify what the 'critical friend' approach is and reference this method of credibility.

We have clarified this important step and inserted a reference: "Credibility strategies used throughout analysis included (1) critical friend approach by encouraging reflexivity by asking challenging and provocative questions to critique our research in a supportive manner that clarifies ideas (e.g. interview questions) and interpretation (e.g. codes, themes), and (2) crosschecking of themes identified in prior focus group studies.[12]"

P18, Discussion. While the aims and methods of the paper are clear and relevant, the paper doesn't quite draw on the significance of the interviewee subjects being essential workers. Is it possible to say a little more in the discussion about the significance of this.

Thank you for highlighting this important aspect of our study. We have added: "A strength of our approach is the focus on essential workers given their repeated exposure to higher-risk settings throughout the pandemic, particularly when stay at home orders were imposed for the rest of society. This allows us to capture unique experiences of essential workers who were at the coalface of dealing with the threat of COVID-19 before widespread vaccination in society."

P18, line 60. "indicating people trust these sources" - this seems a bit of an overstatement. A sample of 18 cannot be generalised to 'people'. P19, line 3. Could you explain more fully by what you mean by: "In some instances, failure to share the best available information led people to seek info from misleading sources." I could not locate this in your results?

We agree with these two comments and have revised this section: "We identified that most participants obtained information from official organisations such as national and international health organisations, indicating that the people interviewed in the current study saw these sources as credible, or at least, worth seeking guidance from. In some instances, failure to share the best available information led people to seek info from misleading sources as described in subthemes 2.2 desire for factual information on local area events, and 3.2 zoning out from formal communications."

P21, line 36-47. You mention the use of loss-framed messages as a means to motivate change in behaviour. However, message framing is a much debated topic. Your discussion would be strengthened here by linking to literature on the use of message framing during Covid-19. We expanded our discussion on this point, in a manner relevant to our findings, adding: "Ultimately, this emphasises the importance of message content (i.e. what to say) and message executions (i.e. how to say it) when communicating information on COVID-19.[37]. The need for reiterative approaches to develop public health communications has been highlighted and largely involves monitoring perceptions of message receivers of classic message inputs such as who, says what, through which channel, to whom, and with what effect.[37, 38] However, regardless of the message executions or framing, our study found that interviewed participants were sceptical about the underlying assumptions of message content (e.g. accuracy of information being communicated, or the relevance to their local community)."

Whilst this is an important aspect of our interviews, unfortunately, there was not sufficient material to support a deeper analysis. We believe this could be an important consideration in upcoming research by informing future message framing studies.

Strengths and limitations: Please add that the small sample size means that the results are not generalisable.

We have added: "Additionally, our sample size (n=18) means our results cannot be generalised to the population or workforce as a whole. However, as participants represented essential workers, awareness of their views are important to inform public health policies that cater for cohorts at increased risk of infection."

Response to Reviewer 2:

"Communication, that's the key": A qualitative investigation of how essential workers with COVID-19 responded to public health information

Dear Dr Tmam Al-Ghunaim

We thank you for your constructive comments and thoughts, which have allowed us to further develop our article. Specifically, we thank you for the opportunity to clarify the decision to use qualitative research methods and the interpretation of our findings in the context of other studies.

We hope you find now our manuscript ready for publication in BMJ Open.

Many thanks,

Mark

I have some concerns about the article's first one in the introduction section; it is not clear for the reader what the aim of this study is? What does this study try to solve? We made an error in submitting an unfinished sentence in our manuscript. This has been corrected to, and re-emphasised at the start of our discussion.

We also added: "This research could inform workplace-related public health communications as the current pandemic continues and future pandemics emerge."

Also, I have some concerns about using qualitative methods; why did you choose the interview rather than collecting data by the survey?

Qualitative inquiry is useful to explain the complex, real-world phenomenon (Bradley, Curry, & Devers, 2007), in this instance, how essential workers at increased risk of COVID-19 infections sought and processed public health information. The use of a survey would have required that we restrict answers to pre-specified fields. This approach would have had two major limitations. First, it assumes we already know most of what is worth knowing about the the experiences of essential workers. Second, it deprives us of the opportunity to discover new information important for catering communication strategies for the target group (i.e. essential workers).

In the method section, who is the target sample? I think this also you need to make it clear in the title.

Thank you for the opportunity to clarify our target sample. Our methods sections now confirms that our participants were "a sub-group of the population known as 'essential workers' that could not avail of 'remote working' throughout the pandemic. We have also revised our title based on your suggestion: "Communication, that's the key": A qualitative investigation of how essential workers with COVID-19 responded to public health information.

We need to clarify the meaning of the theme before going to the subtheme. As a reader, it is unclear what you mean by barriers to ongoing engagement. Also, what is the link between the theme and the aim of the study? I think you need to link them in the result section.

We agree and have now added an explanation of each theme in the results section. We have also outlined the link between our aim which relates to re-forming opinions of COVID-19 as well as the government and officials responding to it.

VERSION 2 – REVIEW

REVIEWER	Al-Ghunaim, Tmam University of Leeds, school of psychology
REVIEW RETURNED	15-Apr-2022

GENERAL COMMENTS	This paper uses a standard interview analysis technique to examine how essential employees with confirmed illnesses reacted to COVID-19 information. There are some minor comments can improve this paper: Regarding the method section, why are you choosing the thematic analysis technique, specifically? Also, the authors mentioned that we targeted this group given their first-hand experience of responses to minimize transmission; as a reader, it is not clear what you mean by that? I think this paper needs a conclusion section that summarizes the study.
---

	The paper is well written and well referenced. This study will add to the growing literature about the COVID-19 pandemic.
--	---

VERSION 2 – AUTHOR RESPONSE

Response to Reviewer 2:

“Communication, that’s the key”: A qualitative investigation of how essential workers with COVID-19 responded to public health information

Dear Ms Tmam Al-Ghunaim

On behalf of the authors I wish to extend my gratitude for taking the time to review our manuscript and making suggestions to clarify our findings. We appreciate your positive feedback on the importance of our findings, that it is well written and references, and will add to the growing literature about the COVID-19 pandemic.

We hope you that find now our manuscript ready for publication in BMJ Open.

Many thanks,

Mark

This paper uses a standard interview analysis technique to examine how essential employees with confirmed illnesses reacted to COVID-19 information. There are some minor comments can improve this paper: Regarding the method section, why are you choosing the thematic analysis technique, specifically?

Although thematic analysis is a standard and common qualitative research method we added:

“We used a thematic analysis approach to support a structured exploration of patterns in perceived behaviours and contextual factors associated with COVID-19 outbreaks experienced by the participants. As qualitative research seeks to generate knowledge grounded in human experience, thematic analysis is a trusted method that guides researchers in ‘highlighting similarities and differences’, ‘exploring unanticipated insights’ and patterns, and ‘summarising key features’ of data that cannot be statistically analysed.[9] Failure to overlook the important of qualitative methods diminishes our understanding of how people experience specific events and how that experience might be enhanced through new considerations when designing interventions (e.g. infection control measures in high-risk workplaces).”

Also, the authors mentioned that we targeted this group given their first-hand experience of responses to minimize transmission; as a reader, it is not clear what you mean by that? We added:

All individuals were identified as part of suspected workplace outbreaks where ≥ 2 individuals had epidemiologically linked infections. We targeted this group given their first-hand experience of (1) COVID-19 outbreaks in the workplace, and (2) responses to minimise transmission such as infection control measures in workplaces, engagement with test-and-trace services, and quarantining.

I think this paper needs a conclusion section that summarizes the study.

To our knowledge - which has been confirmed online and through correspondent with the editorial office - BMJ Open does not structure articles like. Hence, we reported summary points in abstract and in bullet points before the main text as per guidelines in original submission and R1.